# Noise-free Loss Gradients: A Surprisingly Effective Baseline for Coreset Selection

**Saumyaranjan Mohanty**                                                                 *ai23resch04001@iith.ac.in*
*Department of Artificial Intelligence*
*Indian Institute of Technology Hyderabad*

**Chimata Anudeep**                                                                      *chanudeep03@gmail.com*
*Department of Computer Science*
*BITS Pilani KK Birla Goa Campus*

**Konda Reddy Mopuri**                                                                   *krmopuri@ai.iith.ac.in*
*Department of Artificial Intelligence*
*Indian Institute of Technology Hyderabad*

**Reviewed on OpenReview:** *https://openreview.net/forum?id=OE4P1tW8iQ*

## Abstract

The exponential rise in size and complexity of deep learning models and datasets have resulted in a considerable demand for computational resources. Coreset selection is one of the methods to alleviate this rising demand. The goal is to select a subset from a large dataset to train a model that performs almost at par with the one trained on the large dataset while reducing computational time and resource requirements. Existing approaches either attempt to identify remarkable samples (e.g., Forgetting, Adversarial Deepfool, EL2N, etc.) that stand out from the rest or solve complex optimization (e.g., submodular maximization, OMP) problems to compose the coresets. This paper proposes a novel and intuitive approach to efficiently select a coreset based on the similarity of loss gradients. Our method works on the hypothesis that gradients of samples belonging to a given class will point in similar directions during the early training phase. Samples with most neighbours that produce similar gradient directions, in other words, that produce noise-free gradients, will represent that class. Through extensive experimentation, we have demonstrated the effectiveness of our approach in out-performing state-of-the-art coreset selection algorithms on a range of benchmark datasets from CIFAR-10 to ImageNet with architectures of varied complexity (ResNet-18, ResNet-50, VGG-16, ViT). We have also demonstrated the effectiveness of our approach in Generative Modelling by implementing coreset selection to reduce training time for various GAN models (DCGAN, MSGAN, SAGAN, SNGAN) for different datasets (CIFAR-10, CIFAR-100, Tiny ImageNet) while not impacting the performance metrics significantly. Source code is provided at URL.

## 1 Introduction

Large-scale datasets have become essential for training state-of-the-art deep learning models. All the application fronts of Artificial Intelligence, such as Computer Vision, Natural Language, and Speech Processing, have accumulated massive datasets. Concurrently, the complexity of deep learning models has grown steadily. Modern deep learning tasks (especially supervised ones) require extensive hyperparameter tuning during training to achieve the best performance. These have resulted in an exponential rise in computational requirement (Schwartz et al., 2019) and carbon footprint(Heikkilä; Killamsetty et al., 2021a).

Coreset selection aims to mitigate the above issues by finding the most representative data samples from the given training set. Mainly, coreset selection attempts to approximate the learning characteristics of the complete data (e.g., the loss function) (Feldman, 2020). By obtaining a representative coreset with a cardinality of a fraction of the entire dataset, training duration and computational requirement for end-to-end training can be reduced significantly while delivering the desired generalization performance.

Forgetting(Toneva et al., 2019) attempts to identify data points that are difficult to learn as the coreset. The idea presented is that a model trained on "hard to classify" samples naturally generalizes onto the simpler samples. Similarly, training loss based methods attempt to select samples that contribute more to the training of deep neural networks. Gradient Norm (GraNd) and the Error L2-Norm (EL2N) scores introduced by Paul et al. (2023) prune significant fractions of training data without sacrificing much of the classifier's generalization performance. Section 2 comprehensively summarizes the existing works.

Multiple lines of thought based on loss gradients have emerged to compose coresets. Gradient matching based methods aim to compose a coreset whose gradients (weighted combination) closely approximate the gradients produced by the entire training dataset. For instance, CRAIG (Mirzasoleiman et al., 2020) suggests finding the optimal coreset by converting the gradient matching problem to a submodular function maximization and using a greedy approach to optimize it. Because of the individual weights (or learning rates) on the samples in CRAIG, it may not be straightforward to train the models on the coresets in the mini-batch gradient descent framework. Hence, the training speed gains are not maximal on such coresets.

Similarly, Gradmatch(Killamsetty et al., 2021a) constructs an objective for matching the gradients computed over the coreset with that calculated over the complete dataset. They propose minimizing the matching error by casting the objective as a weakly submodular maximization problem and solving it using an orthogonal matching pursuit (OMP) based greedy algorithm. As it identifies the coreset by selecting different subsets during model training to perform gradient matching, training must be carried out each time to obtain coresets at various fractions of the entire dataset.

In this paper, we propose a novel methodology for selecting a coreset based on the hypothesis that most samples from the same class produce gradients in similar directions (particularly in the early stages of the model training, elaborated further in Section 3, Figures 1 and 2). Our method proposes that samples with a substantial gradient similarity to many others belonging to the same class would be ideal candidates for constituting the representative coreset. This approach differs from the existing coreset ideas driven by 'distinctness' or 'difficulty', which prefer samples that give rise to noisy (or minority) loss gradients. Hence, we refer to the proposed method as 'Noise-free Gradients' for coreset selection. In the case of a multi-class classification task, we identify representative samples class-wise and combine them into a coreset.

Our approach uses cosine similarity between gradient vectors to rank data samples according to their ability to represent other samples. Based on the desired coreset size, the top-ranked samples from each class are picked to represent the whole dataset. This makes the proposed approach computationally attractive compared to the optimization-driven coreset ideas (Refer to Section 2 for a detailed review of such works). Moreover, while the existing approaches, such as CRAIG, use a per-sample learning rate while training from the coreset, our method uses single learning rate for all the samples. We also introduce various steps to reduce the computational time. We have applied this coreset selection methodology to two popular and diverse application areas, namely, 1) Image classification and 2) Generative Modelling. The details are discussed in Section 4.

In summary, the major contributions of our work can be summarized as:

- Contrary to the existing 'difficulty/distinctness' notions, this paper introduces a novel, intuitive coreset selection method for identifying class-wise representative samples driven by noise-free loss gradients.

- Also, compared to the existing gradient-based coreset selection methods that are computationally demanding (e.g., submodular function optimization), our method is computationally efficient.

- We thoroughly evaluate the proposed method over multiple object recognition datasets of varying complexity. Experiments with different deep neural network classifiers (CNN and Vision Trans-

former) and cross-architecture generalization studies demonstrate that our method achieves higher accuracy than state-of-the-art coreset selection methods.

- We apply the proposed method in the context of GAN training and demonstrate consistent performance improvement over existing methods.

## 2 Related Works

Multiple coreset selection methods have recently been proposed. Here, we briefly present the most prominent works.

Toneva et al. (2019) proposed coreset selection through catastrophic forgetting. During the training, they defined a flip in classifying a training example from the correct label to an incorrect one as a 'forgetting' event. Removing unforgettable examples results in minimal performance drop while speeding up the training and hyperparameter tuning.

K-Center Greedy approximation (Sener & Savarese, 2018) attempts to solve the minimax facility location problem to select coresets from a large dataset such that the maximum distance between points in the non-coreset and its closest point in the coreset is minimized.

Uncertainty-based methods work on the idea that samples having lower confidence may have a higher impact during training than those with higher confidence. Thus, these methods suggest constituting the coresets with the samples with lower confidence. Commonly used metrics to calculate sample uncertainty are least confidence (Shen et al., 2018), entropy (Settles, 2012), and margin (Coleman et al., 2020).

Adversarial Deepfool (Ducoffe & Precioso, 2018) and Contrastive Active Learning (Margatina et al., 2021) work to find data points distributed near the decision boundary. Samples that lie close to the classification boundary are treated as difficult samples to learn. Error-based methods try to select samples that contribute more to the training of the neural networks. Two metrics called the Gradient Normed (GraNd) and the Error L2-Norm (EL2N) scores are introduced by Paul et al. (2023) that help in pruning significant fractions of training data without sacrificing test accuracy. *GraNd* measures the importance of each sample to the training loss at early epochs. *EL2N* approximates the *GraNd* score, which measures the norm of the error vector, with a higher score indicating higher potential influence. The authors conclude that the images with higher scores tend to be harder to learn (forgettable examples). Their method chooses these forgettable samples as the coreset.

RETRIEVE (Killamsetty et al., 2021d) formulates coreset selection for Semi-Supervised Learning (SSL) as a bi-level optimization problem. This method considers labelled and unlabeled sets to develop the bi-level optimization problem. It uses a greedy algorithm to select the coreset that minimizes the labelled set loss. GLISTER (Killamsetty et al., 2021b), Generalization-based Data Subset Selection for Efficient and Robust Learning, applies bi-level optimization for supervised and active learning. It formulates coreset selection as a bi-level optimization problem that maximizes the log-likelihood on a held-out validation dataset.

Gradient matching-based methods work on the expectation that the optimal coreset can approximate the gradients produced by the entire training dataset. CRAIG (Mirzasoleiman et al., 2020) selects representative subsets that closely approximate the entire gradient. They achieve this by converting the gradient matching problem to optimizing a submodular function using a greedy approach. *Gradmatch* (Killamsetty et al., 2021a) method follows a similar approach. It introduces a squared $L_2$ regularization term over the weight vectors and uses a greedy Orthogonal Matching Pursuit (OMP) algorithm to select the coreset iteratively. After training on the resulting coreset for a pre-determined number of epochs, the algorithm repeats the coreset construction using the latest iterate.

Several methods have utilized submodular functions (Liu, 2020) to leverage their ability to measure diversity and information, then use greedy algorithms to maximize a submodular function for coreset selection. Iyer et al. (2021) have studied various submodular functions such as Graph Cut, entropy, and facility location and utilize greedy optimization algorithms for data summarization.

Xia et al. (2023) have proposed a universal method termed 'Moderate Coreset' to tackle the task-specific nature of dataset selection. They calculate the distance between the hidden representation of samples and the representational class centres. Based on these Euclidean distances, they rank the data points in an ascending order and select the data points closest to the distance median as a coreset.

Yang et al. (2023) proposed a parameter influence-based optimization for dataset selection. They argue that not all training data contributes uniformly to the model's parameter learning. So, an influence function-based iterative method can be designed to linearly estimate the impact of omitting a specific subset on the model parameters. However, they have not considered lower coreset sizes than 50% of the original dataset size.

Sinha et al. (2019) introduced the application of coreset selection to sub-sample a larger batch to produce a smaller batch to speed up GAN training significantly while not sacrificing performance to a significant level. DeVries et al. (2020) introduced instance selection based upon the high-density region of the data manifold to speed up GAN training.

Guo et al. (2022) has developed an excellent and comprehensive code library named *DeepCore* that implements current popular and state-of-the-art coreset selection methods in a unified framework based on PyTorch (Paszke et al., 2019). DeepCore framework has provided a unified comparative analysis among 15 algorithms for CIFAR-10 and 14 algorithms for ImageNet-1K. The algorithms compared are: Contextual Diversity (Agarwal et al., 2020), Herding (Welling, 2012), k-Center Greedy (Sener & Savarese, 2018), Least Confidence, Entropy, Margin (Coleman et al., 2020), Forgetting (Toneva et al., 2019), GraNd (Paul et al., 2023), Cal (Margatina et al., 2021), DeepFool (Ducoffe & Precioso, 2018), Craig (Mirzasoleiman et al., 2020), GradMatch (Killamsetty et al., 2021b), Glister (Killamsetty et al., 2021b), Facility Location and GraphCut (Iyer et al., 2021). Due to its large running time, the DeepFool algorithm is not considered for the ImageNet-1K dataset. We have also utilized publicly available implementation of Moderate Coreset (Xia et al., 2023) for CIFAR-10, CIFAR-100 and Tiny ImageNet datasets and presented the comparative analysis in Section 5.

## 3 Methodology

This section provides a detailed description of the proposed *Noise-free Gradients* approach. We first focus on the supervised classification task of object recognition using Deep Neural Network classifiers (CNNs and Vision Transformers) because of their proven effectiveness. We have also applied our proposed method to speed up GAN training without sacrificing the performance of the model significantly. Table 1 provides the notation used throughout the paper.

Table 1: Notation

| Symbol | Description |
|---|---|
| $\mathbb{V} = \{(x_i, y_i)\}$ | Large Training Dataset |
| $\mathbb{S} \subset \mathbb{V}$ | Coreset of $\mathbb{V}$ (target) |
| $\theta$ | Parameters of the classifier |
| $\Phi$ | Threshold on the gradient similarity for neighborhood identification |
| $\mathbb{V}_c$ | Training data belonging to class c |
| $g_{x_i}^{\theta}$ | Loss gradients computed for data sample $(x_i, y_i)$ at the last fully connected layer |
| $\|x\|$ | $l_2$ norm of $x$ |
| $\|x\|_1$ | $l_1$ norm of $x$ |
| $|\mathbb{A}|$ | Cardinality of set $\mathbb{A}$ |
| $\mathbb{1}$ | Indicator function |
| $< ., . >$ | Dot product operator |

Our objective is to select a representative subset $\mathbb{S}$ of the complete dataset $\mathbb{V}$ such that model $\theta^{\mathbb{S}}$ trained on $\mathbb{S}$ has a generalization performance close to that of the model $\theta^{\mathbb{V}}$ trained on $\mathbb{V}$. Training deep neural networks

is reduced to an empirical risk minimization problem often optimized in the gradient descent framework. In practice, the incremental Gradient (IG) methods, such as Stochastic Gradient Descent (SGD), iteratively estimate the gradient on mini-batches of training data that construct the parameter updates.

Existing gradient-based coreset selection methods such as CRAIG(Mirzasoleiman et al., 2020) and Gradmatch(Killamsetty et al., 2021a) try to find an optimal coreset such that the weighted sum of the gradients of the coreset elements remains within an error margin of the gradients of the entire dataset. The objective function can be written as:

$$\underset{w, \mathbb{S}}{\arg\min} \, F\left(\frac{1}{|\mathbb{V}|} \sum_{(x_i,y_i)\in \mathbb{V}} g_{x_i}^\theta, \frac{1}{\|w\|_1} \sum_{(x_i,y_i)\in \mathbb{S}} w_{x_i} g_{x_i}^\theta\right) \tag{1}$$

Where $w$ is the vector of weights associated with the elements of the coreset, and $F$ is a distance metric.

It can be re-written as:

$$\mathbb{E}_\theta\left[\frac{1}{|\mathbb{V}|} \sum_{(x_i,y_i)\in \mathbb{V}} g_{x_i}^\theta\right] = \mathbb{E}_\theta\left[\frac{1}{\|w\|_1} \sum_{(x_i,y_i)\in \mathbb{S}} w_{x_i} g_{x_i}^\theta\right] + \epsilon \tag{2}$$

Where $\epsilon$ is the error term having the same dimension as the gradient vector, note that equation 2 considers the expected value of the approximation error in the parameter space.

Our method identifies the representative samples based on their gradients. Intuitively, data samples whose gradients are similar to most other samples best approximate the complete dataset. In other words, these representative samples result in local minima close to the minima achieved by the entire dataset.

Our method thus selects a subset $\mathbb{S}$ of a desired cardinality that closely approximates the whole dataset $\mathbb{V}$. We define the normalized gradient similarity between two samples as

$$\rho(x_i, y_i, x_j, y_j, \theta) = \frac{< g_{x_i}^\theta, g_{x_j}^\theta >}{\|g_{x_i}^\theta\|\|g_{x_j}^\theta\|} \tag{3}$$

For each sample $x_i$ in the dataset $\mathbb{V}$, we can measure its ability to represent $\mathbb{V}$ as

$$f(x_i) = \mathbb{E}_\theta\left[\sum_{x_j\in\mathbb{V}, \, j\neq i} \rho(x_i, y_i, x_j, y_j, \theta)\right] \tag{4}$$

Using this measure, we compute the suitability of every sample in $\mathbb{V}$ to become an element of the coreset $\mathbb{S}$. Essentially, this translates to sorting the dataset samples in the decreasing order of this measure and composing the coreset of a desired cardinality. The exact steps involved in this process are described in the next paragraph.

We start with a randomly initialized model and update its parameters on the complete dataset for a few epochs. In our experiments, we observe that generally, 5 to 10 epochs (during which the loss value doesn't plateau) are sufficient. We save the checkpoints of the model parameters after each epoch. For each of these checkpoints, we calculate the gradients of the loss function with respect to the model parameters computed at each data sample. We score the dataset samples based on their ability to represent other samples as denoted in equation 4. We aggregate these scores across the saved checkpoints as an approximation to the expectation over the parameter space.

Figure 1 shows the histogram of pairwise cosine similarity of gradients of images belonging to airplane class for CIFAR-10 dataset on ResNet-18 architecture for initial four epochs. As we can see, during initial training, the most images have gradients with high cosine similarity values, indicating moving in similar directions. Figure 2 shows the histogram of pairwise cosine similarity of gradients of images belonging to (i) airplane

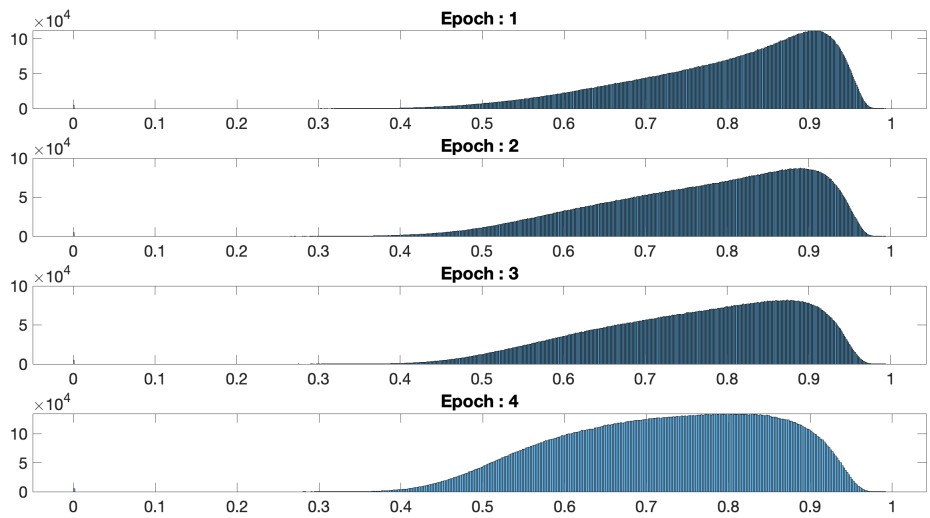

Figure 1: Histogram of pairwise cosine similarity of gradients during the training of ResNet-18 for the initial four epochs. Samples belong to the 'airplane' class of CIFAR-10 dataset. A strong similarity (or alignment) of the gradients can be observed.

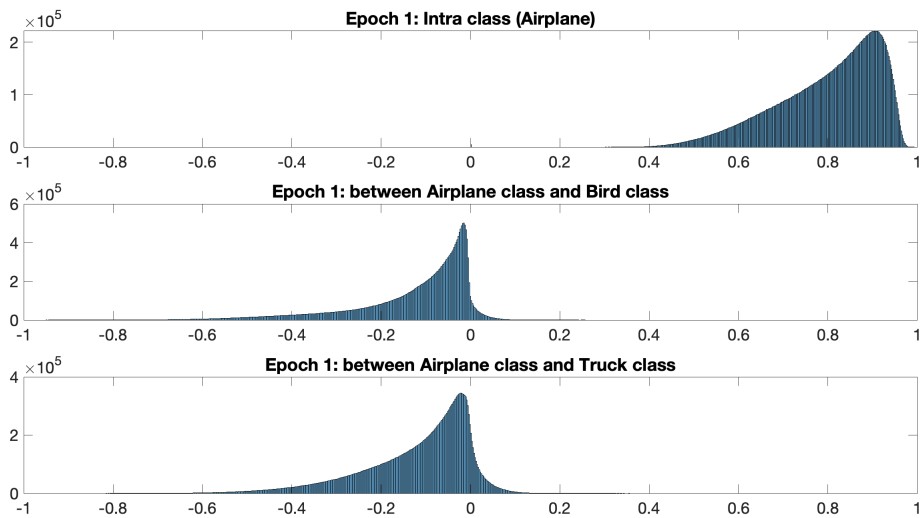

Figure 2: histogram of pairwise cosine similarity of gradients of images belonging to (i) airplane class (top panel), (ii) airplane and bird class (middle panel), and (iii) airplane and truck class (bottom panel). While intra-class samples display strong similarity, inter-class samples have gradients in different directions.

class (top panel), (ii) airplane and bird class (middle panel), and (iii) airplane and truck class (bottom panel). All the computations are done for the model saved after the first epoch. It indicates that images belonging to the same class will have gradients in similar directions, while images belonging to different classes will have gradients in different directions. This behaviour forms the basis of our method for computation of the coreset.

To improve the time efficiency of our approach, we slightly modify it to utilize a nearest-neighbor search algorithm. We utilize radius-based neighbour learning to obtain the local density of data points. The per-

sample scores denoted by equation 4 are measured in terms of the number of dataset samples present nearby in the gradient space. A nearest neighbour search algorithm finds the number of samples within a given radius from each sample (in other words, with similarity more than a threshold $\Phi$) and assigns it as its score. We then rank the samples according to their aggregated scores across multiple checkpoints. The top-ranked images are selected as the representative coreset.

The formulation of *Noise-free Gradients* can be represented as shown in equation 5. For a given class $c$, the top-ranked samples $x_j$ are selected based on their scores.

$$x_j = \underset{(x_i, y_i) \in \mathbb{V}_c}{\arg\max} \sum_{\theta} \sum_{j \neq i} \mathbb{1}(\rho(x_i, y_i, x_j, y_j, \theta) > \Phi) \tag{5}$$

Algorithm 1 presents our approach more formally.

---

**Algorithm 1** Noise-free Gradients for Coreset algorithm

---

**Require:** Train set: $\mathbb{V}$; Total epochs: $T$; number of classes: $C$; number of coreset images per class: $N$;
 Model checkpoint after initial T epochs : $\theta$
**Ensure:** Coreset $\mathbb{S}$
1: **for** class $c$ in 1,..., $C$ **do**
2:  **for** $(x_i, y_i) \in \mathbb{V}_c$ **do**
3:   **for** epochs $t$ in 1,..., $T$ **do**
4:    compute $g_{x_i}^{\theta_t}$
5:   **end for**
6:   $f(x_i) = \sum_{\theta_t} \sum_{(x_j, y_j) \in \mathbb{V}_c, j \neq i} \mathbb{1}(\rho(x_i, y_i, x_j, y_j, \theta_t) > \Phi)$
7:   Store $f(x_i)$
8:  **end for**
9: **end for**
10: $\mathbb{S} = \emptyset$
11: **for** class $c$ in 1,..., $C$ **do**
12:  $\mathbb{S} \leftarrow \mathbb{S} \cup \operatorname{argsort}_{(x_i, y_i) \in \mathbb{V}_c} f(x_i)[: N]$          ▷ Descending order
13: **end for**

---

## 4 Implementation of Noise-free Gradients

We have implemented the proposed method *Noise-free Gradients* using the PyTorch framework.

### 4.1 Gradients with respect to the Classification layer

Deep learning models such as ResNet (He et al., 2015; 2016), VGG (Simonyan & Zisserman, 2015), and ViT Small (Lee et al., 2021) have millions of parameters, and it is impractical to consider the gradient of loss function with respect to each of the parameters. It is observed that the gradient mostly captures the variation of gradient norm with respect to the parameters of the last (classification) layer of the neural network (Katharopoulos & Fleuret, 2019). Similar to the earlier works (Ash et al., 2020; Killamsetty et al., 2021a), we avoid computation of the gradient of the loss function with respect to all the model parameters, restricting to only the final fully connected layer while calculating the score of data samples (equation 4).

### 4.2 Nearest Neighbor algorithm

The complexity of similarity score computation (equation 4) among the samples of a particular class of size N is $\mathcal{O}(N^2)$. The complexity and computation time are linear in the number of classes in the dataset $C$. Instead of computing $N$ similarity scores for each sample in a given class, our approach finds the number of these $N$ samples within a close neighborhood. This vital modification saves nontrivial complexity. We have utilized the radius-based nearest neighbor algorithm implementation from scikit-learn (Pedregosa et al.,

2011) library with the proposed distance (or similarity) measure mentioned in equation 3 as the metric for identification of neighbors within a given radius. Pairwise cosine similarity function of torchmetrics library[1] is used. This results in a $\approx 30X$ speed up of the computation time compared to the naive method. We have used a threshold value ($\Phi$) of 0.2 in all our experiments. For the ablation experiments on the effect of this threshold, readers can refer to the supplementary document (Section 2).

## 5 Experiments and Results

### 5.1 Experimental Setup

**Applications**. We have evaluated our coreset selection methodology on image classification and GAN training.

**Datasets**. For image classification application, we evaluate the effectiveness of our method on four popularly used benchmark object recognition datasets, i.e., CIFAR-10(Krizhevsky, 2009), CIFAR-100(Krizhevsky, 2009), Tiny ImageNet (Le & Yang, 2015) and ImageNet-1K (Russakovsky et al., 2015). CIFAR-10 dataset consists of $50,000$ colour images of dimension $32 \times 32 \times 3$ from 10 different classes, each class having $5,000$ images. CIFAR-100 dataset consists of $50,000$ training images from 100 classes with 500 training images per class. Tiny ImageNet dataset consists of $100,000$ training images from 200 classes with 500 training images per class. ImageNet-1K is a subset of the larger dataset ImageNet, an image dataset organized according to the WordNet hierarchy. ImageNet-1K consists of 1000 classes, with $1,281,167$ training images and $50,000$ validation images. In Summary, to test the robustness of our method, we have considered four popular classification datasets with the number of classes varying from 10 to 1000.

**Classifier**. We have used randomly initialized ResNet-18 (He et al., 2016), ResNet-50 (He et al., 2015), VGG-16 (Simonyan & Zisserman, 2015) and ViT Small (Lee et al., 2021) architectures to carry out comparative performance analysis of existing SOTA algorithms against our method. The first three are CNN architectures that vary in complexity and number of parameters, and the fourth is a vision transformer. We have also carried out cross-architecture generalization to demonstrate the effectiveness of our method.

**GAN models**. For GAN training application, we evaluate the effectiveness of our method on four different GAN architectures i.e. DCGAN (Goodfellow et al., 2014), MSGAN (Mao et al., 2019), SNGAN Miyato et al. (2018) and SAGAN Zhang et al. (2019).

**GAN Metrics**. We have considered the Inception Score (IS) (Salimans et al., 2016) and Fretchet Inception Distance (FID) (Heusel et al., 2018) for comparing the performance of our method against existing coreset selection for GAN methodologies.

**Baselines** We have considered the results reported by DeepCore benchmark wherever available and utilized DeepCore implementation to carry out experiments wherever results are not available.[2] We have utilized publicly available implementation of Moderate-DS[3] and instance selection for GAN[4]. Abbreviations used in results for different methods which are part of DeepCore are: GC(GraphCut), F(Forgetting), R(Random), FL(Facility Location), GN(GraNd), CR(CRAIG), H(Herding).

**Implementation** Experiments for CIFAR-10 and CIFAR-100 are carried out for ten individual runs with different random seeds. Experiments for tiny ImageNet were carried out for five separate runs, and ImageNet1K were carried out for two individual runs due to substantial computational requirements. In each iteration, number of training epochs were set at 200, and an SGD optimizer with a momentum of 0.9 was used with a batch size of 128 and a learning rate of 0.1. For ImageNet-1K, batch size of 256 was used.

For GAN training, we have used default settings for all the GAN models recommended by their respective authors. For instance selection, a 50% retention ratio is used. For our method, a 50% coreset selection percentage is used. FID scores are lower the better, and Inception scores are higher the better. The number of steps used for full dataset training is 200k, and the steps used for coreset-based training is 100k.

---

[1]`https://lightning.ai/docs/torchmetrics/stable/pairwise/cosine_similarity.html`
[2]`https://github.com/PatrickZH/DeepCore`
[3]`https://github.com/tmllab/2023_ICLR_Moderate-DS`
[4]`https://github.com/uoguelph-mlrg/instance_selection_for_gans`

## 5.2 Results for Image Classification

### 5.2.1 Results for CIFAR-10

A comparison of accuracy values obtained from DeepCore, Moderate-DS, and *Noise-free Gradients* on the CIFAR-10 dataset for various fractions of the original dataset is presented in Table 2, Table 3, Table 4 and Table 5 for ResNet-18, ResNet-50, VGG-16 and ViT Small architecture respectively. The algorithm with the best accuracy at each percentage level reported by DeepCore is mentioned within brackets against the accuracy value. The average rank of a method is calculated as the mean of its rank across different fractions of the dataset considered. As can be observed from the values, our method outperforms all the existing SOTA algorithms across architectures of diverse complexity and depth.

Table 2: Comparison of results for CIFAR-10 on ResNet-18

| Size | DeepCore | Moderate-DS | Our Method |
|---|---|---|---|
| 0.5% | 34.90 ± 2.30 (GC) | 33.93 ± 1.10 | **43.64 ± 0.87** |
| 1.0% | 42.80 ± 1.30 (GC) | 39.83 ± 0.56 | **54.06 ± 0.92** |
| 5.0% | 65.70 ± 1.20 (GC) | 69.20 ± 1.30 | **72.96 ± 0.28** |
| 10.0% | 76.60 ± 1.50 (GC) | 79.10 ± 0.12 | **79.12 ± 0.40** |
| 20.0% | 87.10 ± 0.50 (R) | 86.49 ± 0.29 | **87.18 ± 0.45** |
| 30.0% | **91.70 ± 0.3 (F)** | 89.39 ± 0.12 | 89.62 ± 0.50 |
| **Rank** | 2.17 | 2.67 | **1.17** |

Table 3: Comparison of results for CIFAR-10 on ResNet-50

| Size | DeepCore | Moderate-DS | Our Method |
|---|---|---|---|
| 0.5% | 32.85 ± 1.20 (GC) | 13.73 ± 4.82 | **47.32 ± 0.69** |
| 1.0% | 36.58 ± 1.50 (GC) | 18.86 ± 1.55 | **56.06 ± 0.10** |
| 5.0% | 63.31 ± 1.53 (GC) | 53.89 ± 1.03 | **73.04 ± 0.32** |
| 10.0% | 70.64 ± 1.10 (F) | 65.47 ± 1.83 | **78.06 ± 0.41** |
| 20.0% | 85.06 ± 1.48 (FL) | 84.00 ±0.79 | **86.11 ± 0.48** |
| 30.0% | **89.83 ± 0.30 (F)** | 88.74 ± 0.27 | 89.22 ± 0.20 |
| **Rank** | 1.83 | 3.00 | **1.17** |

Table 4: Comparison of results for CIFAR-10 on VGG-16

| Size | DeepCore | Moderate-DS | Our method |
|---|---|---|---|
| 0.5% | 32.85 ± 1.2 (GC) | 13.73 ±4.82 | **47.32 ± 0.69** |
| 1.0% | 36.58 ± 1.5 (GC) | 18.86 ±1.55 | **56.06 ± 0.10** |
| 5.0% | 63.31 ± 1.53 (GC) | 53.89 ±1.03 | **73.04 ± 0.32** |
| 10.0% | 70.64 ± 1.1 (F) | 65.47 ±1.83 | **78.06 ± 0.41** |
| 20.0% | 85.06 ± 1.48 (FL) | 84.00 ±0.79 | **86.11 ± 0.48** |
| 30.0% | **89.83 ±0.3 (F)** | 88.74 ±0.27 | 89.22 ±0.2 |
| **Rank** | 1.67 | 3 | **1.33** |

Table 5: Comparison of results for CIFAR-10 on ViT small

| Size | DeepCore | Moderate-DS | Our Method |
|---|---|---|---|
| 0.5% | 30.05 ± 1.21 (GC) | 23.10 ± 1.41 | **35.22 ± 0.31** |
| 1.0% | 37.31 ± 0.52 (GC) | 27.00 ± 4.80 | **38.50 ± 0.27** |
| 5.0% | 53.96 ± 0.45 (GC) | 51.30 ± 2.30 | **56.82 ± 0.20** |
| 10.0% | 57.43 ± 0.69 (GC) | 54.40 ± 2.35 | **58.32 ± 0.16** |
| 20.0% | 64.98 ± 0.90 (GC) | 66.10 ± 1.66 | **67.16 ± 0.52** |
| 30.0% | 67.38 ± 1.01 (GC) | 70.40 ± 0.28 | **71.43 ± 0.49** |
| 40.0% | 71.24 ± 0.20 (GC) | 72.60 ± 1.40 | **75.45 ± 0.70** |
| 50.0% | 72.98 ± 2.10 (F) | 73.10 ± 4.09 | **78.07 ± 0.68** |
| 60.0% | **77.68 ± 0.20 (GC)** | 76.95 ± 1.85 | 77.11 ± 0.10 |
| **Rank** | 2.33 | 2.55 | **1.11** |

### 5.2.2 Results for CIFAR-100

A comparison of accuracy values obtained from DeepCore, Moderate-DS, and *Noise-free Gradients* on the CIFAR-100 dataset for various fractions of the original dataset is presented in Table 6, Table 7, Table 8 and Table 9 for ResNet-18, ResNet-50, VGG-16 and ViT Small architectures respectively. The algorithm with the best accuracy at each percentage level reported by DeepCore is mentioned within brackets against the accuracy value. The average rank of a method is calculated as the mean of its rank across different fractions of the dataset considered. We outperform all the other methods for training on ResNet-50, VGG-16 and ViT Small architecture while performing better for lower selection percentages (up to 20%) for training on ResNet-18 architecture.

### 5.2.3 Results for Tiny ImageNet and ImageNet-1K

We believe the effectiveness of coreset selection algorithms must be tested in the face of complex datasets. Hence, we evaluate our algorithm with the Tiny ImageNet and ImageNet-1K datasets. Similar to other experiments, we train multiple models on the resulting coresets. Each model is trained for 200 epochs with a random PyTorch seed. The comparative analysis is tabulated in Table 10, 11, 12 and 13. The algorithms with the best accuracy at each percentage level reported by DeepCore are mentioned within brackets against the accuracy value. The average rank of a method is calculated as the mean of its rank across different fractions of the dataset considered. Our method consistently outperforms all the other coreset selection algorithms.

Table 6: Comparison of results for CIFAR-100 on ResNet-18

| Size | DeepCore | Moderate-DS | Our Method |
|------|----------|-------------|------------|
| 0.5% | 8.49 ± 0.68 (GC) | 5.36 ±0.12 | **12.0 ± 0.11** |
| 1.0% | 13.04 ± 0.22 (GC) | 7.78 ± 0.20 | **18.95 ± 0.08** |
| 5.0% | 31.18 ± 0.37 (GC) | 18.62 ± 0.33 | **37.67 ± 0.10** |
| 10.0% | 41.98 ± 0.88 (GC) | 32.57 ± 1.40 | **46.48 ± 0.28** |
| 20.0% | 56.68 ± 0.98 (GC) | 54.4 ± 0.19 | **58.06 ± 0.34** |
| 30.0% | **64.05 ± 0.28 (GC)** | 62.97 ± 0.41 | 62.58 ± 0.24 |
| 40.0% | **69.37 ± 0.30 (GC)** | 67.33 ± 0.22 | 66.31 ± 0.16 |
| 60.0% | **73.80 ± 0.31 (GN)** | 72.02 ± 0.30 | 70.74 ± 0.20 |
| **Rank** | **1.63** | 2.63 | 1.75 |

Table 7: Comparison of results for CIFAR-100 on ResNet-50

| Size | DeepCore | Moderate-DS | Our Method |
|------|----------|-------------|------------|
| 0.5% | 7.34 ± 0.62 (GC) | 3.64 ±0.62 | **10.61 ± 0.22** |
| 1.0% | 12.69 ± 0.28 (GC) | 5.48 ±0.42 | **15.85 ± 0.16** |
| 5.0% | 25.7 ± 0.55 (GC) | 14.70 ±0.24 | **31.64 ± 0.19** |
| 10.0% | 36.91 ± 0.18 (GC) | 22.74 ±1.67 | **41.32 ± 0.48** |
| 20.0% | 48.14 ± 0.23 (GC) | 51.83 ±0.52 | **54.18 ± 0.51** |
| 30.0% | 56.1 ± 0.32 (GC) | 57.79 ±1.61 | **60.82 ± 0.08** |
| 40.0% | 62.52 ± 0.23 (GC) | 64.92 ±0.93 | **65.01 ± 0.24** |
| 50.0% | 68.62 ± 1.20 (GC) | **69.11 ±0.62** | 68.67 ± 0.17 |
| 60.0% | **73.34 ± 0.39 (F)** | 71.87 ±0.91 | 71.10 ± 0.38 |
| **Rank** | 2.33 | 2.33 | **1.33** |

Table 8: Comparison of results for CIFAR-100 on VGG-16

| Size | DeepCore | Moderate-DS | Our Method |
|------|----------|-------------|------------|
| 0.5% | 2.45 ± 0.57 (GC) | 2.00 ± 0.10 | **8.92 ± 0.45** |
| 1.0% | 4.47 ± 1.28 (GC) | 2.10 ± 0.23 | **15.2 ± 0.46** |
| 5.0% | 27.23 ± 0.70 (GC) | 13.70 ± 0.91 | **32.12 ± 0.58** |
| 10.0% | 38.17 ± 1.32 (GC) | 28.10 ± 1.60 | **44.67 ± 0.05** |
| 20.0% | 52.23 ± 0.47 (GC) | 49.90 ± 0.30 | **53.95 ± 0.30** |
| 30.0% | 58.87 ± 0.45 (GC) | 59.50 ± 0.30 | **61.23 ± 0.17** |
| 40.0% | 62.52 ± 0.23 (GC) | 64.92 ± 0.93 | **66.17 ± 0.23** |
| 50.0% | 68.62 ± 1.21 (GC) | 69.11 ± 0.62 | **69.47 ± 0.19** |
| 60.0% | **73.34 ± 0.39 (F)** | 71.87 ± 0.91 | 71.82 ± 0.20 |
| **Rank** | 2.22 | 2.55 | **1.22** |

Table 9: Comparison of results for CIFAR-100 on ViT Small

| Size | DeepCore | Moderate-DS | Our Method |
|------|----------|-------------|------------|
| 0.5% | 7.45 ± 0.17 (GC) | 5.20 ± 0.06 | **11.03 ± 0.23** |
| 1.0% | **14.57 ± 0.47 (GC)** | 7.82 ± 0.20 | 14.35 ± 0.18 |
| 5.0% | 24.50 ± 0.71 (GC) | 21.12 ± 0.30 | **24.85 ± 0.30** |
| 10.0% | 32.00 ± 0.34 (GC) | 27.00 ± 0.50 | **32.60 ± 0.30** |
| 20.0% | 40.73 ± 1.19 (GC) | 38.51 ± 0.47 | **41.30 ± 0.80** |
| 30.0% | 45.50 ± 1.01 (GC) | 44.90 ± 0.31 | **48.12 ± 0.48** |
| 40.0% | 46.72 ± 1.02 (GC) | 46.20 ± 0.60 | **51.75 ± 2.23** |
| 50.0% | 47.11 ± 2.00 (GC) | 52.83 ± 0.32 | **57.62 ± 0.64** |
| 60.0% | 49.66 ± 0.64 (GC) | 55.00 ± 0.40 | **60.81 ± 0.53** |
| **Rank** | 2.11 | 2.77 | **1.11** |

Table 10: Comparison of results for Tiny ImageNet on ResNet-18

| Size | DeepCore | Moderate-DS | Our Method |
|------|----------|-------------|------------|
| 0.5% | 3.29 ± 0.23 (FL) | 3.34 ± 0.33 | **6.91 ± 0.24** |
| 1.0% | 5.34 ± 0.45 (GC) | 4.11 ± 0.24 | **11.09 ± 0.45** |
| 5.0% | 13.73 ± 0.71 (GC) | 14.03 ± 1.01 | **22.83 ± 0.98** |
| 10.0% | 21.23 ± 0.16 (GC) | 21.40 ± 0.41 | **28.05 ± 0.82** |
| 20.0% | 29.97 ± 0.72 (GC) | 30.99 ± 0.93 | **34.91 ± 0.43** |
| 30.0% | 33.73 ± 0.37 (GC) | 37.18 ± 0.87 | **40.36 ± 0.43** |
| 40.0% | 35.74 ± 0.57 (F) | 40.21 ± 0.24 | **40.41 ± 0.17** |
| 60.0% | 39.74 ± 0.73 (F) | **46.19 ± 0.64** | 44.74 ±0.44 |
| **Rank** | 2.75 | 2.12 | **1.12** |

Table 11: Comparison of results for Tiny ImageNet on ResNet-50

| Size | DeepCore | Moderate-DS | Our Method |
|------|----------|-------------|------------|
| 0.5% | 2.26 ± 0.32 (GC) | 1.71 ± 0.27 | **5.24 ± 0.03** |
| 1.0% | 5.36 ± 0.37 (F) | 3.38 ± 0.13 | **10.55 ± 0.41** |
| 5.0% | 20.56 ± 0.33 (GC) | 12.39 ± 0.02 | **26.07 ± 0.19** |
| 10.0% | 29.83 ± 0.49 (GC) | 23.21 ± 0.23 | **34.08 ± 0.21** |
| 20.0% | 39.60 ± 1.32 (GC) | 34.71 ± 0.36 | **43.95 ± 1.03** |
| 30.0% | 45.80 ± 0.23 (F) | 41.65 ± 0.20 | **50.82 ± 0.03** |
| 40.0% | 50.00 ± 1.72 (GC) | 48.72 ± 0.61 | **55.23 ± 0.08** |
| 60.0% | 56.97 ± 0.12 (F) | 56.10 ± 0.13 | **61.38 ±0.11** |
| **Rank** | 2.00 | 3.00 | **1.00** |

Table 12: Comparison of results for Tiny ImageNet on ViT small

| Size | DeepCore | Moderate-DS | Our Method |
|------|----------|-------------|------------|
| 0.5% | 2.88 ± 0.04 (FL) | 3.35 ± 0.20 | **5.92 ± 0.12** |
| 1.0% | 4.23 ± 0.25 (F) | 4.34 ± 0.13 | **9.18 ± 0.25** |
| 5.0% | 8.41 ± 0.15 (GC) | 13.56 ± 0.23 | **18.10 ± 0.15** |
| 10.0% | 11.53 ± 1.23 (GC) | 15.84 ± 0.22 | **22.58 ± 0.06** |
| 20.0% | 12.54 ± 0.06 (GC) | 22.03 ± 0.32 | **28.26± 0.31** |
| 30.0% | 14.09 ± 0.11 (GC) | 25.35 ± 0.31 | **33.66 ± 0.21** |
| 40.0% | 15.69 ± 0.45 (GC) | 28.40 ± 0.22 | **37.29 ± 0.32** |
| 50.0% | 17.34 ± 1.05 (GC) | 30.68 ± 0.14 | **40.92 ± 0.36** |
| 60.0% | 19.23 ± 0.32 (GC) | 32.74 ± 0.27 | **43.80 ±0.38** |
| **Rank** | 3.00 | 2.00 | **1.00** |

Table 13: Comparison of results for ImageNet-1K on ResNet-18

| Size | DeepCore | Noise-free Gradients |
|------|----------|----------------------|
| 0.1% | 1.29 ± 0.09 (CAL) | **1.88 ± 0.10** |
| 0.5% | 7.66 ± 0.43 (GraphCut) | **11.58 ± 0.15** |
| 1.0% | 18.10 ± 0.22 (GraNd) | **22.82 ± 0.07** |
| 5.0% | **47.64 ± 0.03 (Forgetting)** | 43.50 ± 0.41 |
| 10.0% | **55.12 ± 0.13 (Forgetting)** | 49.51 ± 0.26 |
| **Rank** | 1.6 | **1.4** |

We visualize the top and bottom-ranked samples in this subsection according to our coreset selection algorithm. We chose the "Zebra" class from the 1000 ImageNet classes. Figure 3 presents the top-ranked and bottom-ranked 12 images of the "Zebra" class. The selected samples clearly emphasize the fact that the images with top ranks are representative of the class. In contrast, images at the bottom positions are ambiguously labelled (have parts of objects from other classes) or difficult to label as "Zebra". As our objective is to select the most representative images from each class, the visualization provides evidence for the intuition over which the proposed method is founded. We have provided additional examples in the supplementary material.

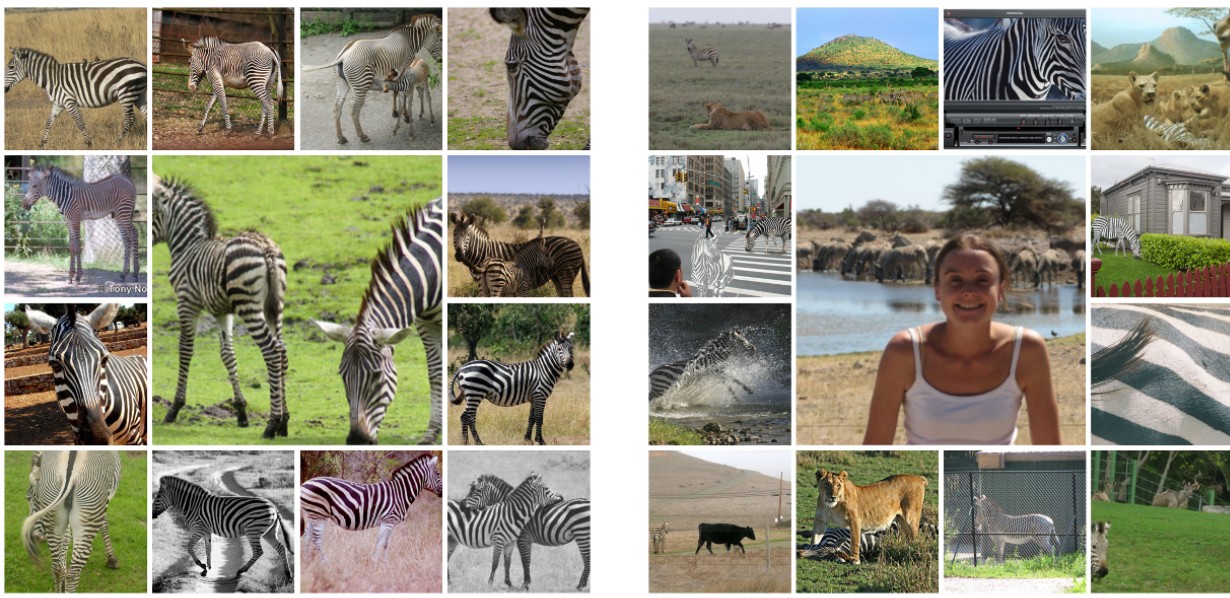

Top ranked images            Bottom ranked images

Figure 3: Top-ranked and bottom-ranked 12 images from the "zebra" class from ImageNet-1K dataset by the proposed "Noise-free Gradients" approach. Images are ranked as per the number of neighbors within a threshold gradient similarity value. Top-ranked images are unambiguous representative of the class, while bottom-ranked images are either mislabeled or ambiguously labeled.

### 5.2.4 Robustness against image noise

In realistic scenarios, training data may be polluted by corrupted images(Xia et al., 2023). We have randomly added five different types of noise to the training subset of the original dataset, namely Gaussian Noise, random occlusion, resolution change, fog and motion blur. We have conducted coreset selection on the corrupted train set and observed its generalization performance on the uncorrupted test set.

We have adapted the implementation provided by Moderate-DS(Xia et al., 2023) available in public github repository.[5] Table 14 compares accuracy obtained by all the methods available in DeepCore library and Moderate with our *Noise-free Gradients* method for a 30% of the training dataset impacted by noise. ResNet-18 architecture is used for evaluation. As can be seen, for this significant amount of noise added to the original dataset, our method outperforms all the other methods for most of the coreset selection. As our proposed method composes the coreset with images having gradient similarity with a higher number of images from the same class, the coreset is able to perform better than existing methods even when the training dataset is corrupted by significant image noise.

Table 14: Comparison of results for CIFAR-100 with 30% corruption on ResNet-18

| Size | DeepCore | Moderate-DS | Our Method |
|------|----------|-------------|------------|
| 0.5% | $6.16 \pm 0.23$ (CR) | $5.07 \pm 0.57$ | $\mathbf{11.1 \pm 0.18}$ |
| 1.0% | $8.74 \pm 0.32$ (CR) | $17.56 \pm 0.61$ | $\mathbf{33.56 \pm 0.65}$ |
| 5.0% | $17.58 \pm 0.36$ (GC) | $19.03 \pm 0.8$ | $\mathbf{23.65 \pm 0.15}$ |
| 10.0% | $20.7 \pm 0.89$ (GC) | $29.20 \pm 0.23$ | $\mathbf{31.24 \pm 0.11}$ |
| 20.0% | $24.40 \pm 0.98$ (GC) | $41.27 \pm 0.56$ | $\mathbf{41.89 \pm 0.40}$ |
| 30.0% | $28.14 \pm 0.76$ (GC) | $\mathbf{57.58 \pm 0.16}$ | $51.93 \pm 0.25$ |
| **Rank** | 2.83 | 2 | **1.17** |

Table 15: Comparison of results for CIFAR-100 with 5% corruption on ResNet-50

| Size | DeepCore | Moderate-DS | Our Method |
|------|----------|-------------|------------|
| 10.0% | $22.2 \pm 0.13$ (H) | $35.1 \pm 0.34$ | $\mathbf{42.0 \pm 0.24}$ |
| 20.0% | $42.50 \pm 1.27$ (H) | $46.78 \pm 1.9$ | $\mathbf{53.48 \pm 0.22}$ |
| 30.0% | $53.88 \pm 0.37$ (H) | $57.36 \pm 1.22$ | $\mathbf{59.9 \pm 0.13}$ |
| 40.0% | $60.54 \pm 0.94$ (H) | $\mathbf{65.4 \pm 0.39}$ | $64.96 \pm 0.2$ |
| 60.0% | $70.22 \pm 0.22$ (F) | $\mathbf{71.46 \pm 0.19}$ | $70.27 \pm 0.16$ |
| **Rank** | 3.00 | 1.60 | **1.40** |

Table 15 and Table 16 show accuracy comparison for 5% and 20% of images from CIFAR-100 dataset impacted by noises, trained on ResNet-50 architecture. Table 17 compares accuracies obtained for 20% noise

---

[5]https://github.com/tmllab/2023_ICLR_Moderate-DS

Table 16: Comparison of results for CIFAR-100 with 20% corruption on ResNet-50

| Size | DeepCore | Moderate-DS | Our Method |
|------|----------|-------------|------------|
| 10.0% | 24.78 ± 0.45 (F) | 27.05 ±0.8 | **42.32 ± 0.23** |
| 20.0% | 44.42 ±0.46 (H) | 42.98 ±0.87 | **54.63 ± 0.26** |
| 30.0% | 53.57 ±0.31 (H) | 55.80 ±0.96 | **60.58 ±0.20** |
| 40.0% | 60.72 ± 1.78 (H) | 61.84 ±1.96 | **64.83 ±0.35** |
| 60.0% | 69.10 ± 1.73 (H) | 70.05 ±1.29 | **70.51 ± 1.31** |
| **Rank** | 2.80 | 2.20 | **1.00** |

Table 17: Comparison of results for CIFAR-100 with 20% corruption on ViT Small

| Size | DeepCore | Moderate-DS | Our Method |
|------|----------|-------------|------------|
| 1.0% | 10.02 ± 0.29 (CR) | 7.20 ±0.31 | **10.44 ± 0.10** |
| 5.0% | 17.58 ± 0.36 (GC) | 19.03 ±0.8 | **23.65 ± 0.15** |
| 10.0% | 20.7 ± 0.89 (GC) | 29.20 ±0.23 | **31.24 ± 0.11** |
| 20.0% | 24.40 ±0.98 (GC) | 41.27 ±0.56 | **41.89 ± 0.40** |
| 30.0% | 28.14 ±0.76 (GC) | **57.58 ±0.16** | 51.93 ±0.25 |
| **Rank** | 2.80 | 2.00 | **1.20** |

ratio in the CIFAR-100 dataset, trained on transformer architecture(ViT Small is used). In all four studies, we can see that our method significantly outperforms all the SOTA methods.

### 5.2.5 Robustness against label noise

Practical datasets may also involve label noise, where some images are mislabeled. We have adapted the implementation provided by Moderate-DS (Xia et al., 2023) available in public GitHub repository[6] to generate mislabeled data. Table 18 and Table 19 compare accuracy obtained by all the methods available in the DeepCore library and Moderate with our *Noise-free Gradients* method for a 20% of the training dataset impacted by label noise on ResNet-50 architecture and ViT Small architecture respectively. As can be seen, for this significant amount of noise added to the original dataset labels, our method outperforms all the other methods. As our proposed method composes the coreset with images having gradient similarity with a higher number of images from the same class, the coreset is able to perform better than existing methods even when the training dataset is corrupted by significant label noise.

Table 18: Comparison of results on CIFAR-100 Dataset with 20% label noise on ResNet-50 architecture

| Size | DeepCore | Moderate-DS | Our Method |
|------|----------|-------------|------------|
| 10.0% | 38.59 ±1.29 (GC) | 25.02 ±0.27 | **40.71 ± 0.34** |
| 20.0% | 47.15 ±1.66 (GC) | 42.75 ±1.20 | **52.99 ± 0.51** |
| 30.0% | 57.04 ±0.88 (GC) | 56.28 ±1.11 | **60.77 ±1.17** |
| **Rank** | 2.00 | 3.00 | **1.00** |

Table 19: Comparison of results on CIFAR-100 Dataset with 20% label noise on ViT Small architecture

| Size | DeepCore | Moderate-DS | Our Method |
|------|----------|-------------|------------|
| 10.0% | 28.66 ±0.52 (GC) | 23.02 ±0.32 | **31.39 ± 0.36** |
| 20.0% | 38.79 ±0.55 (GC) | 34.92 ±0.10 | **36.90 ± 0.21** |
| 30.0% | 40.22 ±0.92 (GC) | 43.91 ±0.28 | **44.71 ±0.07** |
| **Rank** | 2.00 | 3.00 | **1.00** |

### 5.2.6 Impact on class-wise accuracy

We visualize the class-wise accuracy of training a ResNet-18 model on the CIFAR-100 dataset, comparing the performance of the full dataset and coreset selection with 20% of the dataset in Figure 4. We observe a Spearman rank-order correlation coefficient of 0.87, indicating a high correlation between class-wise accuracies obtained with the full dataset and 20% of the full dataset as coreset. It can be seen that our method does not adversely impact any particular class during coreset selection. The coreset composed by our method results in class-wise accuracies strongly correlated to that of the full dataset.

### 5.2.7 Cross-architecture generalization

To study the robustness of our coreset selection method, we have carried out cross-architecture generalization performance comparison at 1% and 10% fractions of CIFAR-10 and Tiny ImageNet for two CNN architectures ResNet-18 (He et al., 2016), VGG-16 (Simonyan & Zisserman, 2015) and one transformer based architecture ViT-Small. The comparative analysis is tabulated in Table 20 and Table 21. The "Source" column denotes the architecture on which the coreset is selected, and the "Target" row indicates the architecture on which its performance is evaluated. As can be observed, our proposed method is able to perform consistently better than existing methods in a cross-architecture setting. Cross-architecture analysis on the Tiny-ImageNet dataset is provided in the supplementary material.

---

[6]https://github.com/tmllab/2023_ICLR_Moderate-DS

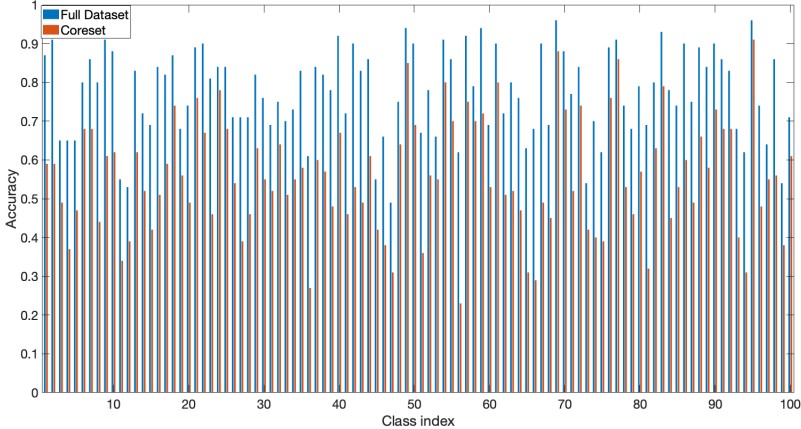

Figure 4: Class-wise accuracy comparison between full dataset and coreset with 20% selection percentage. A high correlation is observed indicating coreset composed by our proposed method not impacting any particular class adversely.

Table 20: Cross-architecture comparison for 1% coreset of CIFAR-10

| Target → | ResNet-18 | | |
|---|---|---|---|
| Source ↓ | DeepCore | Moderate-DS | Our Method |
| ResNet-18 | 42.78 ± 1.30 (GC) | 41.44 ±0.34 | **48.00 ± 2.10** |
| VGG-16 | 43.02 ± 1.30 (GC) | 42.12 ±0.27 | **46.27 ± 0.33** |
| ViT Small | 26.01 ± 2.00 (GC) | 41.76 ±0.35 | **45.05 ± 0.51** |
| Target → | VGG-16 | | |
| Source ↓ | DeepCore | Moderate-DS | Our Method |
| ResNet-18 | 29.01 ± 3.63 (GC) | 44.84 ±0.33 | **47.21 ± 0.95** |
| VGG-16 | 27.47 ± 4.00 (GC) | 44.35 ±0.45 | **47.64 ± 0.71** |
| ViT Small | 35.29 ± 2.82 (GC) | 34.44 ±0.30 | **38.43 ± 0.40** |
| Target → | ViT Small | | |
| Source ↓ | DeepCore | Moderate-DS | Our Method |
| ResNet-18 | 29.06 ± 0.75 (GC) | 34.71 ±0.23 | **40.35 ± 0.90** |
| VGG-16 | 42.06 ± 0.90 (GC) | 31.25 ±0.73 | **44.89 ± 0.55** |
| ViT Small | 22.89 ± 1.45 (GC) | 34.44 ±0.30 | **38.43 ± 0.40** |

Table 21: Cross-architecture comparison for 10% coreset of the CIFAR-10

| Target → | ResNet-18 | | |
|---|---|---|---|
| Source ↓ | DeepCore | Moderate-DS | Our Method |
| ResNet-18 | 76.65 ± 1.48 (GC) | 75.68 ±0.62 | **78.62 ± 1.05** |
| VGG-16 | 78.66 ± 0.55 (GC) | 74.27 ±0.72 | **78.75 ± 0.16** |
| ViT Small | 70.21 ±1.16 (GC) | 73.66 ±0.29 | **75.15 ± 0.42** |
| Target → | VGG-16 | | |
| Source ↓ | DeepCore | Moderate-DS | Our Method |
| ResNet-18 | 75.29 ± 1.05 (GC) | 75.60 ±0.86 | **78.22 ± 0.29** |
| VGG-16 | 77.91 ± 0.71 (GC) | 76.62 ±0.52 | **78.84 ± 0.33** |
| ViT Small | 70.38 ± 0.87 (GC) | 73.81 ±0.68 | **75.47 ± 0.35** |
| Target → | ViT Small | | |
| Source ↓ | DeepCore | Moderate-DS | Our Method |
| ResNet-18 | 62.99 ± 0.12 (GC) | 61.69 ±0.23 | **63.66 ± 0.35** |
| VGG-16 | 63.14 ± 0.18 (GC) | 61.49 ±0.33 | **66.88 ± 0.41** |
| ViT Small | 49.30 ± 1.30 (GC) | 54.40 ±2.35 | **58.32 ± 0.16** |

## 5.3 GAN Analysis

So far, we have presented the effectiveness of our proposed method in the context of training a discriminative model (classifier); however, in this subsection, we investigate the effectiveness of the proposed method in a different context, which is training a generative model. Similar to our approach for the classification task, we train the generative model for the initial few epochs with the full dataset and then select the class-wise coreset with a 50% selection percentage. We utilize the discriminator as a classifier for the purpose of computation of gradient similarity and ranking of images. Thereon, the generative model is trained with the coreset.

### 5.3.1 Results on CIFAR-10

We have trained four different GAN architectures, namely DCGAN, MSGAN, SAGAN, and SNGAN, on the CIFAR-10 dataset. Table 22 compared the FID and IS scores obtained by training with a full dataset and coreset composed by small GAN, instance selection, and our method. As can be observed, our method consistently performs better than the other two methods in the FID score, which is a key metric for measuring the quality of generated images.

Table 22: Comparison of results on CIFAR-10 Dataset. ↓ means lower is better, ↑ means higher is better

| Method | DCGAN | | MSGAN | | SAGAN | | SNGAN | |
|---|---|---|---|---|---|---|---|---|
| | FID ↓ | IS ↑ | FID ↓ | IS ↑ | FID ↓ | IS ↑ | FID ↓ | IS ↑ |
| Full Dataset | 28.80 | 5.84 | 33.16 | 5.81 | 33.73 | 5.56 | 22.44 | 7.36 |
| Small GAN | 35.53 | 5.59 | 45.76 | 5.28 | 44.97 | 4.92 | 35.89 | 6.34 |
| Instance Selection | 31.26 | 5.62 | 39.05 | 5.71 | 36.96 | 5.48 | 27.81 | 6.98 |
| Our Method | 31.14 | 5.70 | 35.65 | 5.51 | 34.60 | 5.42 | 24.46 | 7.22 |

### 5.3.2  Results on CIFAR-100

We have trained four different GAN architectures, namely DCGAN, MSGAN, SAGAN and SNGAN, on the CIFAR-100 dataset. Table 23 compared the FID and IS scores obtained by training with full dataset and coreset composed by small GAN, instance selection and our method. As can be observed, our method consistently performs better than the other two methods in FID score, which is a key metric for measuring the quality of generated images.

Table 23: Comparison of results on CIFAR-100 Dataset. ↓ means lower is better, ↑ means higher is better

| Method | DCGAN | | MSGAN | | SAGAN | | SNGAN | |
|---|---|---|---|---|---|---|---|---|
| | FID ↓ | IS ↑ | FID ↓ | IS ↑ | FID ↓ | IS ↑ | FID ↓ | IS ↑ |
| Full Dataset | 27.98 | 6.13 | 37.59 | 7.26 | 33.37 | 5.79 | 26.97 | 6.26 |
| Small GAN | 32.81 | 6.11 | 71.13 | 6.75 | 41.53 | 5.76 | 39.71 | 5.51 |
| Instance Selection | 30.48 | 6.07 | 53.88 | 7.10 | 39.17 | 5.60 | 28.88 | 6.13 |
| Our Method | 30.08 | 6.08 | 50.22 | 6.64 | 36.15 | 5.61 | 27.71 | 6.14 |

### 5.3.3  Results on Tiny ImageNet

We have trained SAGAN architecture on the Tiny ImageNet dataset. Table 24 compared the FID and IS scores obtained by training with full dataset and coreset composed by small GAN, instance selection and our method.

Table 24: Comparison of results on Tiny ImageNet Dataset. ↓ means lower is better, ↑ means higher is better

| Method | SAGAN | |
|---|---|---|
| | FID ↓ | IS ↑ |
| Full Dataset | 55.67 | 7.12 |
| Small GAN | 75.12 | 6.65 |
| Instance Selection | 69.16 | 6.46 |
| Our Method | 56.81 | 6.97 |

## 5.4  Computation time analysis

### 5.4.1  Image Classification

Our method does not consider all images in a given dataset in a single coreset selection step. Instead, it processes the dataset class-wise, implements multiprocessing and optimizes the nearest neighbour algorithm to parallelize the ranking process. Hence, the methodology can be easily scaled to a large-scale dataset beyond ImageNet-1K. Also, the computation of gradients and nearest neighbours is a one-time process for a given dataset and architecture (per selection epoch). After that, aggregating the individual image ranks across the training epochs takes only 225 seconds (steps 11-13 in Algorithm 1). As we directly utilize the gradients of the loss function with respect to the weights of the last fully connected layer, there is no requirement for storage of the gradient values. Table 25 provides timing analysis for the ImageNet dataset on ResNet-18 architecture on a 48 GB RTX A6000 GPU.

Table 25: Execution time comparison

| Process | Execution Time |
|---|---|
| Single epoch full dataset training | 4.46 hours |
| Gradient Calculation (per epoch) | 3.19 hours |
| Ranking for the coreset | 225 seconds |

### 5.4.2 GAN training

By using 50% of the total dataset, we train the GAN model with coreset selection for half the number of update steps compared to training with the entire dataset. As shown in section 5.3, our method can achieve a comparable FID score to training with a whole dataset while utilizing half of the available dataset. Table 26 provides the total training time required for various methods, and we can see that our method can reduce the total time taken significantly. At the same time, degradation in performance is not significant.

Table 26: Execution time comparison for GAN training on CIFAR-100 Dataset (in hours)

| Method | DCGAN | MSGAN | SAGAN | SNGAN |
|---|---|---|---|---|
| Full Dataset | 2.47 | 5.14 | 2.90 | 9.48 |
| Small GAN | 9.82 | 15.10 | 16.22 | 19.35 |
| Instance Selection | 1.32 | 3.00 | 1.56 | 3.95 |
| Our Method | 1.34 | 3.53 | 1.80 | 3.98 |

## 5.5 Accuracy vs training speed tradeoff

Figure 5 and 6 shows classification accuracy vs computational time trade-off for full dataset training vs coreset training using 10%, 20% and 30% of the entire dataset. As it can be seen, with a small reduction in accuracy, we can improve the training speed, and it shows the effectiveness of the proposed coreset method during hyper-parameter tuning, which may require evaluating different values for them over multiple runs.

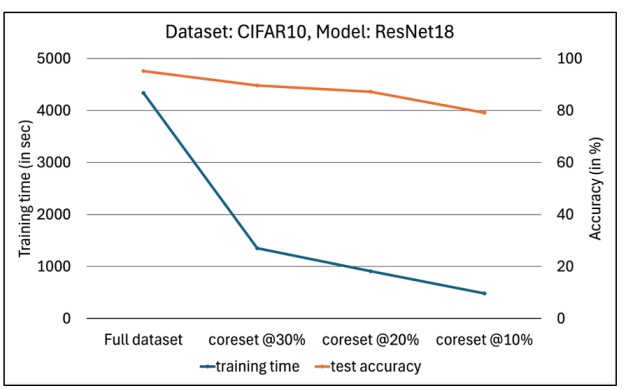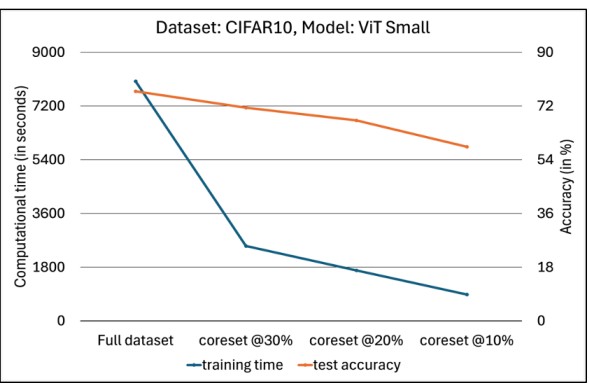

Figure 5: Accuracy vs training speed trade-off. It can be observed that using 30% of coreset results in similar accuracy while the training time is reduced more than 3 times.

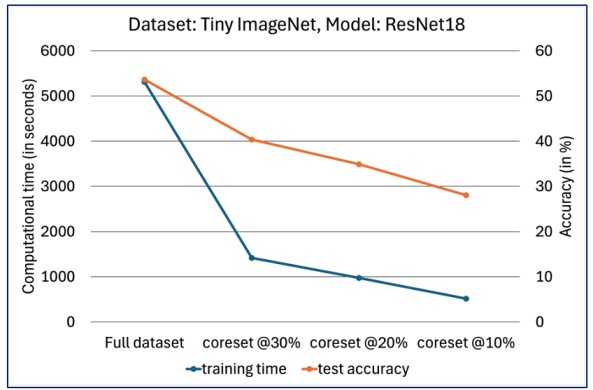 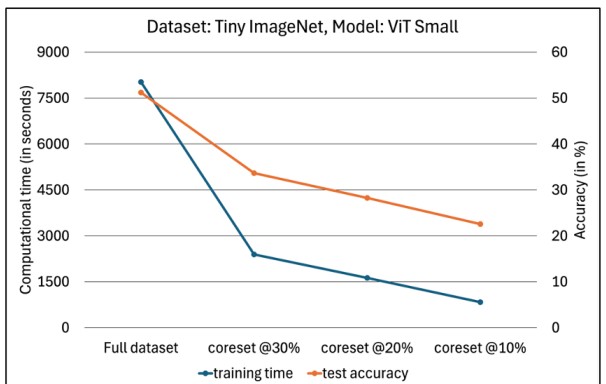

Figure 6: Accuracy vs training speed trade-off. For a complex dataset like Tiny ImageNet, the reduction in training time is not insignificant for applications like hyperparameter tuning where model needs to be trained multiple times.

## 6 Conclusions

In this paper, we introduced *Noise-free Gradients*, an intuition-driven, gradient similarity-based coreset selection method for identifying representative instances from large training datasets. We studied its effectiveness through our extensive experiments on popular object recognition datasets (CIFAR-10, CIFAR-100, Tiny ImageNet, ImageNet-1K) and sophisticated model architectures (CNNs and Vision Transformer) at various coreset sizes. We demonstrated superior generalization performance of classifiers trained on the resulting coresets. We have also demonstrated that our model achieves consistently higher performance in the case of cross-architecture generalization. Thus, we strongly propose our method as an essential baseline to benchmark the forthcoming coreset selection methods.

A branch of existing methods, such as Forgetting (Toneva et al., 2019), utilizes noisy gradients to compose coresets. The argument put forward by these methods is the samples that are hard to learn will be able to provide better generalization on the test set. However, our approach explores an opposing notion. Samples that derive loss gradients similar to many other samples can effectively represent the whole dataset and thus form a coreset. This hypothesis naturally exploits the redundancy in large datasets to compose extremely small coresets. Our method outperforms other coreset selection methods even in the presence of noise in the original dataset. We have also applied our coreset method to improve computational time significantly for GAN training without sacrificing much performance.

### 6.1 Future Directions

While one can appreciate the simplicity and intuition behind the proposed coreset selection method, it must be studied further, particularly concerning the approximation error between the gradients computed by the coreset and the entire dataset. Inducing diversity in the selected samples along with individual importance weights (or per-sample step size) would further improve the effectiveness of the proposed approach. One may consider these aspects for future study.

There have been a few reported applications of coreset selection, namely Active Learning (Sener & Savarese, 2018) and semi-supervised learning (Killamsetty et al., 2021c). It would be interesting to study the effectiveness of our *Noise-free Gradients* approach concerning these application areas.

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
