# 1 Cross Architecture Study

In the main paper, we have provided a cross-architecture study on the CIFAR-10 dataset at 1% and 10% of coreset selection on three different architectures (two CNN-based and one transformer-based). Table 1 and table 2 present cross-architecture classification results for 1% and 10% of the Tiny ImageNet dataset.

Table 1: Cross-architecture comparison for 1% coreset of Tiny ImageNet

| Target → | ResNet-18 | | |
|---|---|---|---|
| Source ↓ | DeepCore | Moderate-DS | Our Method |
| ResNet-18 | 7.10 ± 0.44 (GraphCut) | 4.11 ±0.07 | **11.61 ± 0.24** |
| VGG-16 | 8.41 ± 0.24 (GraphCut) | 4.13 ±0.31 | **11.16 ± 0.30** |
| ViT Small | 9.03 ± 0.17 (GraphCut) | 3.34 ±0.16 | **9.04 ± 0.19** |
| Target → | VGG-16 | | |
| Source ↓ | DeepCore | Moderate-DS | Our Method |
| ResNet-18 | 5.77 ± 0.44 (GraphCut) | 5.11 ±0.05 | **8.61 ± 0.32** |
| VGG-16 | 6.41 ± 0.26 (GraphCut) | 4.20 ±0.10 | **9.32 ± 0.26** |
| ViT Small | 3.80 ± 0.27 (GraphCut) | 3.90 ±0.19 | **6.25 ± 0.49** |
| Target → | ViT Small | | |
| Source ↓ | DeepCore | Moderate-DS | Our Method |
| ResNet-18 | 3.71 ± 0.11 (GraphCut) | 4.88 ±0.08 | **10.36 ± 0.23** |
| VGG-16 | 4.10 ± 0.11 (GraphCut) | 4.73 ±0.21 | **10.17 ± 0.32** |
| ViT Small | 5.13 ± 0.25 (GraphCut) | 4.22 ±0.05 | **8.48 ± 0.14** |

Table 2: Cross-architecture comparison for 10% coreset of Tiny ImageNet

| Target → | ResNet-18 | | |
|---|---|---|---|
| Source ↓ | DeepCore | Moderate-DS | Our Method |
| ResNet-18 | 42.78 ± 1.30 (GraphCut) | 41.44 ±0.34 | **48.00 ± 2.10** |
| VGG-16 | 43.02 ± 1.30 (GraphCut) | 42.12 ±0.27 | **46.27 ± 0.33** |
| ViT Small | 26.01 ± 2.00 (GraphCut) | 41.76 ±0.35 | **45.05 ± 0.51** |
| Target → | VGG-16 | | |
| Source ↓ | DeepCore | Moderate-DS | Our Method |
| ResNet-18 | 29.01 ± 3.63 (GraphCut) | 44.84 ±0.33 | **47.21 ± 0.95** |
| VGG-16 | 27.47 ± 4.00 (GraphCut) | 44.35 ±0.45 | **47.64 ± 0.71** |
| ViT Small | 35.29 ± 2.82 (GraphCut) | 34.44 ±0.30 | **38.43 ± 0.40** |
| Target → | ViT Small | | |
| Source ↓ | DeepCore | Moderate-DS | Our Method |
| ResNet-18 | 29.06 ± 0.75 (GraphCut) | 34.71 ±0.23 | **40.35 ± 0.90** |
| VGG-16 | 42.06 ± 0.90 (GraphCut) | 31.25 ±0.73 | **44.89 ± 0.55** |
| ViT Small | 22.89 ± 1.45 (GraphCut) | 34.44 ±0.30 | **38.43 ± 0.40** |

# 2 Ablation study on threshold value

Figure 1 shows test set accuracy obtained by a model trained on various percentages of coreset selected with different threshold values for gradient similarity

for CIFAR100 dataset with ResNet18 architecture. Figure 2 shows test set accuracy obtained by a model trained on various percentages of coreset selected with different threshold values for gradient similarity for Tiny ImageNet dataset with ResNet18 architecture.

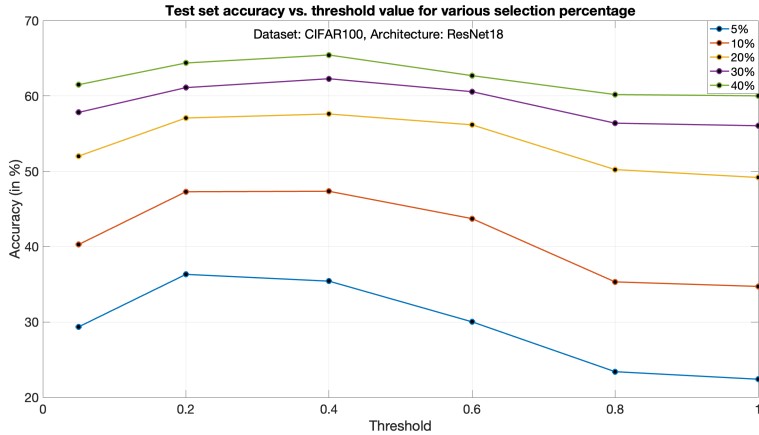

Figure 1: Test set accuracy vs. threshold value for various selection percentages for CIFAR100 dataset with ResNet18 architecture

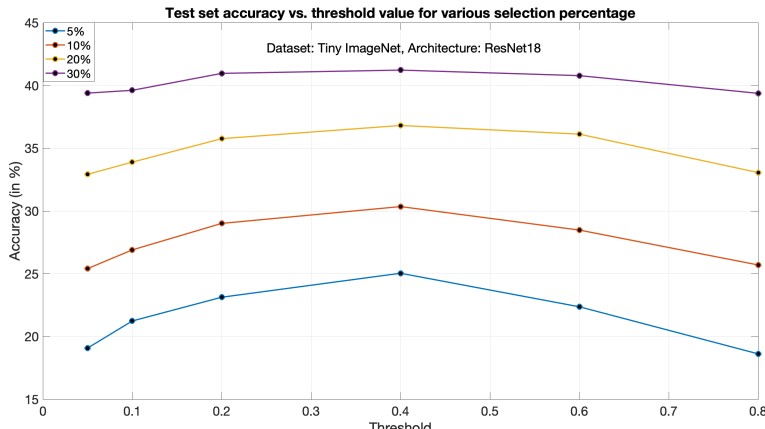

Figure 2: Test set accuracy vs. threshold value for various selection percentages for Tiny ImageNet dataset with ResNet18 architecture

As can be observed from these studies, accuracy value decreases at very low or very high thresholds, with optimum accuracy obtained with a threshold in the range of [0.2,0.4].

# 3 Robustness against image noise

Table 3 tabulates classification accuracies obtained by various methods on Tiny ImageNet dataset with 30% noise on ResNet50 architecture.

Table 3: Comparison of results on Tiny ImageNet Dataset with 30% corruption on ResNet50 architecture

| Percent | DeepCore | Moderate-DS | Our Method |
|---------|----------|-------------|------------|
| 0.5% | 2.16 ± 0.33 (GC) | 1.47 ±0.13 | **2.56 ± 0.28** |
| 1.0% | 5.68 ± 0.45 (GC) | 2.41 ±0.34 | **6.9 ± 0.81** |
| 5.0% | 17.58 ± 0.36 (GC) | 19.03 ±0.8 | **23.65 ± 0.15** |
| 10.0% | 20.7 ± 0.89 (GC) | 29.20 ±0.23 | **31.24 ± 0.11** |
| 20.0% | 24.40 ±0.98 (GC) | 41.27 ±0.56 | **41.89 ± 0.40** |
| 30.0% | 28.14 ±0.76 (GC) | **57.58 ±0.16** | 51.93 ±0.25 |
| **Rank** | 2.83 | 2 | **1.17** |

# 4 Visualization

We visualize a number of classes from the ImageNet-1K dataset with top-ranked and bottom-ranked samples as per our *Noise-free gradients* approach.

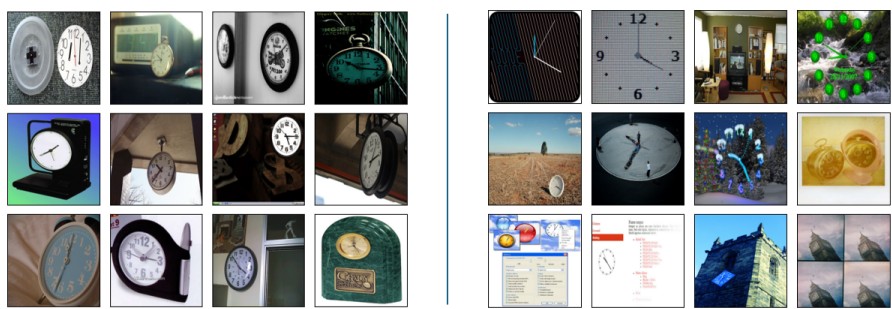

Top ranked images                    Bottom ranked images

Figure 3: Top-ranked 12 images and bottom-ranked 12 images from the 'Analog Clock' class from ImageNet-1K dataset by the proposed 'Noise-free Gradients' approach.

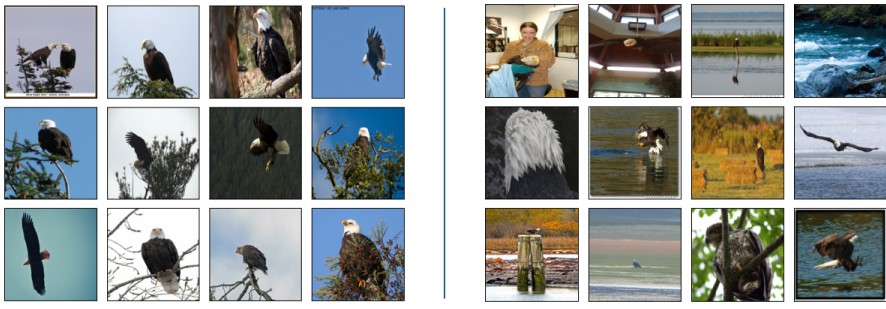

Top ranked images                    Bottom ranked images

Figure 4: Top-ranked 12 images and bottom-ranked 12 images from the 'Bald Eagle' class from ImageNet-1K dataset by the proposed 'Noise-free Gradients' approach.

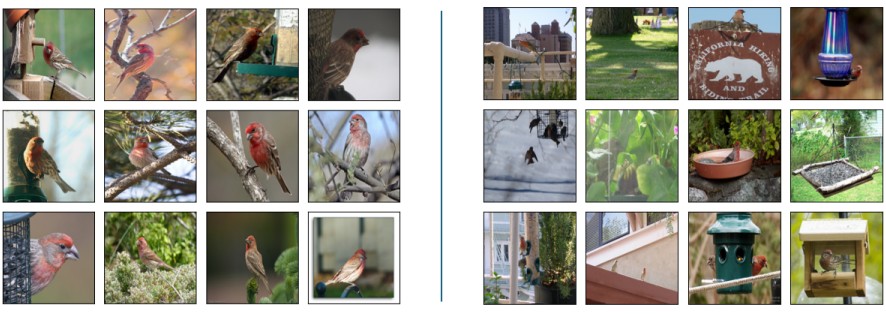

Top ranked images                    Bottom ranked images

Figure 5: Top-ranked 12 images and bottom-ranked 12 images from the 'House Finch' class from ImageNet-1K dataset by the proposed 'Noise-free Gradients' approach.

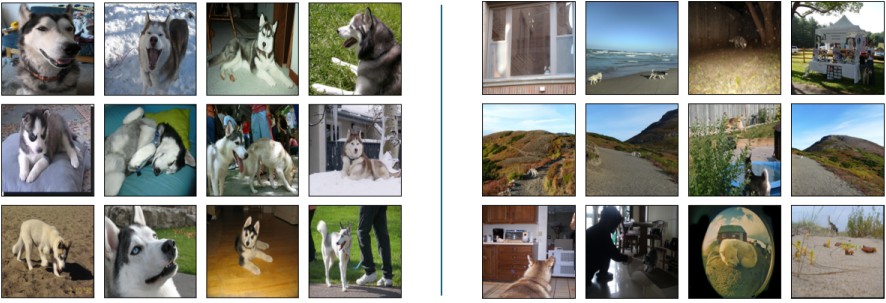

Top ranked images          Bottom ranked images

Figure 6: Top-ranked 12 images and bottom-ranked 12 images from the 'Siberian Husky' class from ImageNet-1K dataset by the proposed 'Noise-free Gradients' approach.

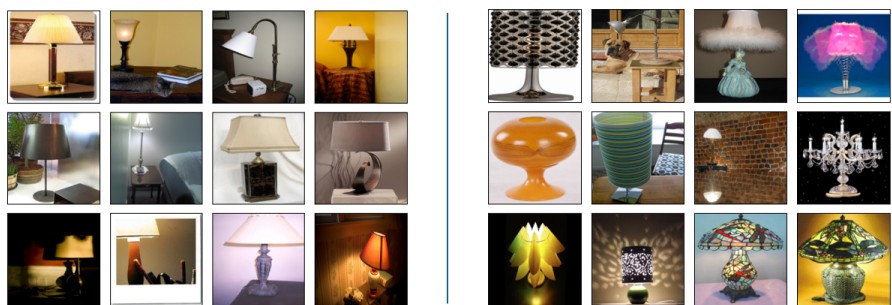

Top ranked images          Bottom ranked images

Figure 7: Top-ranked 12 images and bottom-ranked 12 images from the 'Table Lamp' class from ImageNet-1K dataset by the proposed 'Noise-free Gradients' approach.

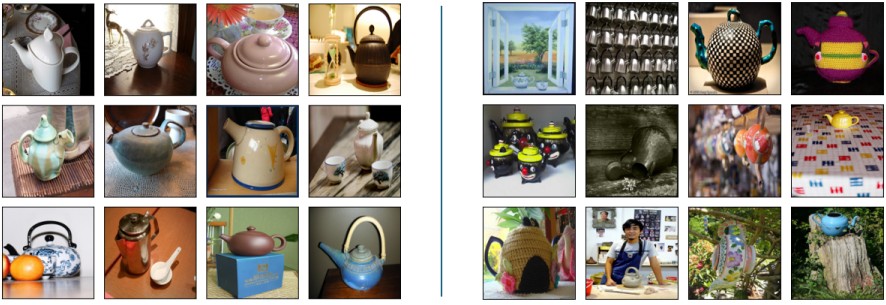

Top ranked images                    Bottom ranked images

Figure 8: Top-ranked 12 images and bottom-ranked 12 images from the 'Tea pot' class from ImageNet-1K dataset by the proposed 'Noise-free Gradients' approach.

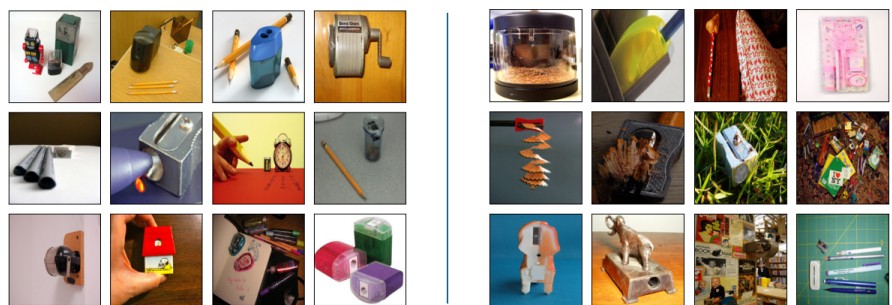

Top ranked images                    Bottom ranked images

Figure 9: Top-ranked 12 images and bottom-ranked 12 images from the 'Pencil Sharpener' class from ImageNet-1K dataset by the proposed 'Noise-free Gradients' approach.