# OpenReview forum: "Noise-free Loss Gradients: A Surprisingly Effective Baseline for Coreset Selection"
_TMLR — Accepted by TMLR_

### Review · Reviewer_R3WH · 2024-10-11

**Summary Of Contributions:**

The paper presents a $\textit{Noise-free Gradients}$ approach for coreset selection. The method addresses the challenge of reducing training computational time and resource requirements by selecting a subset from a larger dataset to train models without hindering the training quality. The core idea of the method is to rely on the similarity between gradients of the training loss during the early training phase.

First, the authors show experimentally that gradients of samples belonging to the same class have a high similarity (measured in terms of cosine similarity) while samples from different classes are not well aligned. The authors then propose a $\textit{Noise-free Gradients}$ algorithm that builds on this idea by first computing the loss gradients for all the samples and then computing a similarity measure that assesses the ability of each sample to represent the other samples in the same class using the cosine similarity above-mentioned.  The final selected subset of size $N$ aggregates the $N$ individual samples that have the highest similarity measure.

The authors propose two implementation simplifications to reduce the computational cost (namely computing the gradients only w.r.t. the classification layer and computing the similarity measure using the nearest neighbors algorithm to avoid calculating cosine similarity between gradients for all the samples of a given class). The authors demonstrate the effectiveness of their approach with extensive experiments for image classification and to speed up Generative Adversarial networks (GAN) training.

**Audience:**

Yes

**Broader Impact Concerns:**

I do not have any ethical concerns.

**Claims And Evidence:**

Yes

**Requested Changes:**

Overall the paper is interesting with an intuitive and efficient approach for coreset selection. The experiments are extensive and showcase significant improvement. The implementation details are given and the authors provide an open-source implementation, which I think is an additional strength of the proposal.

I would like to recommend acceptance for this work.

The changes proposed below would strengthen the work in my opinion. Regarding the weaknesses, could the authors provide
1) Additional ablation on the benefits of nearest neighbors (decrease of computation time v.s. performance compared to computing the gradients on all the samples of a given class)?
2) Additional ablation on the threshold value for the $\textit{Noisy-free Gradients}$ algorithm to compute the similarity measure of each training sample?
3) The anonymized Github link given by the authors has expired by the time of the review. Could the authors make sure that a working link is provided either for camera-ready if the paper is accepted or for a potential resubmission? Indeed, I believe that open-sourcing the code is a strength of this work and could benefit the community.

**Strengths And Weaknesses:**

**Strengths**
- The paper is well written, with clear notations, and motivations;
- The proposed approach is simple yet intuitive;
- The authors proposed an efficient practical implementation to reduce the computational cost;
- The choice of baselines is comprehensive and covers well the literature;
- The experiments are large-scale and enough ablation is provided;
- The authors provide an open-source implementation;
- The authors show significant improvement both for image classification and speeding GAN training.

**Weaknesses**
- The authors could provide additional ablation on the benefits of nearest neighbors (decrease of computation time v.s. performance compared to computing the gradients on all the samples of a given class);
- The authors could provide additional ablation on the threshold value for the $\textit{Noisy-free Gradients}$ algorithm to compute the similarity measure of each training sample.

---

> ### Author Response · Authors · 2024-10-15
> **Response to reviewer R3WH**
>
> We thank the reviewer for their valuable time and feedback. Here are the point-wise responses to the concerns mentioned by the reviewer.
>
> **1.Benefits of nearest neighbors**: The proposed algorithm computes the loss gradients for all the samples belonging to a class. Then, it uses nearest neighbor search to obtain radius-based neighbors for each sample. Note that the nearest neighbor computation we perform is not an approximation algorithm; instead, it is a faster implementation than the naive approach (implementing nested loops). At this stage, we have the ability of each sample to represent other samples in that class. We then consider each class's top-ranked samples to form the coreset. Note that irrespective of the desired coreset size, the loss gradients and nearest neighbors must be computed for all the data samples, which is a one-time process. The gain in the compute results from training the model on the significantly smaller coreset. Presuming the training time is linear in the training dataset size, the coreset size also indicates the required training time. Section 5.5 of the main draft presents the training time vs. accuracy trade-off.
>
>
> **2.Ablation study of threshold value for noise-free gradients algorithm**: We have investigated the impact of threshold on two datasets, namely CIFAR100 and Tiny ImageNet with ResNet18 architecture. We have varied the threshold to calculate the test accuracy of the model trained on the coreset obtained at each threshold for various coreset sizes. Each experiment is repeated 10 times, and the mean accuracy, along with the standard deviation, is reported in the following table. The supplementary document is updated with the plots of accuracy vs. threshold at various selection percentages (Section 2 of the updated supplementary document).
>
> For CIFAR100 dataset:
>
> | Threshold ↓| Selection Percentage→ | 5% | 10% | 20% | 30% | 40% |
> |---|---|---|---|---|---|---|
> |0.05||29.35(1.18)|40.28(1.26)|52.02(1.06)|57.82(0.58)|61.50(0.58)|
> |0.2||36.32(1.00)|47.29(0.68)|57.08(0.70)|61.11(0.84)|64.38(0.39)|
> |0.4||35.42(0.84)|47.35(0.70)|57.61(0.78)|62.29(0.46)|65.44(0.63)|
> |0.6||30.02(1.47)|43.72(1.41)|56.17(0.57)|60.57(0.86)|62.70(0.92)|
> |0.8||23.4(0.68)|35.32(0.94)|50.23(0.50)|56.39(0.77)|60.19(0.71)|
> |1.0||22.4(0.94)|34.72(2.01)|49.18(0.88)|56.05(0.57)|60.02(0.46)|
>
> For Tiny ImageNet Dataset:
> | Threshold ↓| Selection Percentage→  | 5% | 10% | 20% | 30% |
> |---|---|---|---|---|---|
> |0.05||19.08(0.19)|25.41(0.36)|32.92(0.62)|39.39(0.20)|
> |0.1||21.24(0.26)|26.90(0.13)|33.89(0.35)|39.61(0.27)|
> |0.2||23.13(0.29)|29.01(0.02)|35.76(0.20)|40.96(0.30)|
> |0.4||25.04(0.07)|30.35(0.22)|36.81(0.55)|41.22(0.05)|
> |0.6||22.37(0.16)|28.48(0.20)|36.12(0.11)|41.01(0.22)|
> |0.8||18.62(0.07)|25.70(0.33)|33.06(0.26)|39.37(0.15)|
>
> As can be observed from these studies, accuracy value decreases at very low or very high thresholds, with optimum accuracy obtained with a threshold in the range of [0.2,0.4].
>
>
>
> 3. **Link to the anonymized code**: We regret the inconvenience caused by the broken link. The main draft has been updated with the working anonymized link for the repository of our algorithm.

---

> > ### Comment · Reviewer_R3WH · 2024-10-15
> > **Official Comment by Reviewer R3WH**
> >
> > I thank the authors for their prompt answers.
> >
> > 1) Thanks for the explanations regarding the benefits of the nearest neighbors approach, it is clearer now.
> > 2) The ablation study on the threshold seems convincing. I did not find a mention in the main paper of the final value chosen (it seems to be 0.2 according to the code), maybe it is mentioned and I did not see it. In all cases, could the authors simply mention it in Section 4.2 (for instance)?
> > 3) Thanks for the anonymized link, now it works well.
> >
> > I do not have any more comments regarding the submission and maintain my initial recommendation of acceptance.

---

> > > ### Author Response · Authors · 2024-10-16
> > > **Thanks for recommending acceptance**
> > >
> > > We thank the reviewer for recommending to accept our draft. We have revised section 4.2 to mention the threshold value.

---

### Review · Reviewer_EDLA · 2024-11-06

**Summary Of Contributions:**

This submission proposes a new method for coreset selection in deep learning. Instead of running an entire training process on a full dataset which may become prohibitively inefficient/expensive, the goal of coreset selection is to pick a small representative subset of data (i.e. coreset) with the hope that training only on the coreset can achieve comparable accuracy with training on the full dataset.

Many previous coreset selection approaches used the idea of selecting data points that are distinct/hard to learn. The method proposed in this submission is based on a different intuition of selecting data points with high similarity to other data points. Specifically, within each label class, the proposed method selects the N data points with the most neighbors, where the neighbors of a given data point z are other data points within the class whose gradients are close to the gradient at z.

The authors show an efficient way of implementing the proposed coreset selection method based on nearest neighbor search. The authors also perform extensive experiments on image classification and GAN training to demonstrate the effectiveness of the proposed approach. They also test the robustness provided by the proposed coreset selection method when the training data is corrupted.

**Audience:**

Yes

**Broader Impact Concerns:**

I don't see any major broader impact concerns caused by the submission.

**Claims And Evidence:**

Yes

**Requested Changes:**

**Minor**

- Second bullet point on Page 1: demanind -> demanding
- In Figures 5 and 6: I wonder if it is better if the x-axis is arranged in decreasing order: Full dataset, coreset @ 30%, coreset @ 20%, coreset @ 10%.

**Strengths And Weaknesses:**

**Strengths**

- The proposed method is very intuitive, simple, and easy to implement. The computational overhead of constructing the coreset is low. This is shown in the experiments in Sec 5.4.

- Despite the simplicity, the proposed method is very effective. Experiments in the submission show that it achieves better accuracy (for the same coreset size) than previous methods on a variety of tasks (image classification and GAN training) across different datasets. It significantly reduces the computation time needed for training while maintaining good accuracy (Sec 5.5).

- As an extra benefit, the proposed method achieves good robustness when a small part of the dataset is corrupted. A similar idea for achieving robustness has been explored in [this paper](https://arxiv.org/abs/2003.10647).

**Weaknesses**

- The authors claim that their methodology of using gradient similarity to build coresets is novel. However, based on my understanding of the related work section, the Moderate Coreset by Xiao et al. (2017) is also similarity-based, though their similarity is computed on hidden representations rather than gradients. It would be great if the authors could include a more detailed comparison with their work in the introduction, and explain the intuition behind choosing gradients rather than hidden representations to evaluate similarity.

- The authors use the cosine similarities among the gradients to choose the coreset. My understanding is that the length/norm of each gradient is not taken into account when calculating the cosine similarity. That is, the cosine similarities only depend on the directions of the gradients and not on the lengths of the gradients. I'm not sure how this will affect the execution of SGD with momentum (the optimization algorithm used by the authors). (My understanding is that the lengths of the gradients would affect the result of running SGD with momentum.) It would be great if the authors could clarify this and justify the design choice of ignoring the gradient lengths when forming the coreset.

- The proposed method chooses the same number N of data points from each class. I'm wondering what is the idea behind this design choice. Does this lead to better class-wise balance/fairness when the original dataset is imbalanced across label classes?

---

> ### Author Response · Authors · 2024-11-08
> **Response to Reviewer EDLA**
>
> We thank the reviewer EDLA for their valuable time and feedback. here are the point-wise responses to the concerns raised by the reviewer.
>
> **W1**: The Moderate method (Xia et al. 2023) projects the images into the embedding space of the last linear layer before the classification layer. Then, it computes the class centres in that space.  Finally, the images lying within the median radius from the class centre are considered as the coreset to represent the dataset. In our method, we work with the loss gradient similarity to compose the coreset. It is based on the intuition that the direction of the gradient vector during the initial training phase will be similar for the images of a given class (please refer to Figures 1 and 2 in the draft).
>
>
> Note that during the initial training phase, due to the unfinished (far from mature) parameters, the feature space may not cluster the images of the same class closer. Hence, it may not be the right space to identify the representative samples. On the other hand, loss gradients are an excellent tool for achieving this (figures 1 and 2 in the main draft). That way, we can quickly (during the initial epochs itself) identify the representative samples that will compose the coreset.
>
>
>
> **W2**. Our method does not utilize SGD for coreset selection (please refer to Algorithm 1 of the main draft). Only the (plain) gradient vectors are needed to compute the similarity and identify the representative samples. We use vanilla SGD to update the model parameters during the initial training phase (for coreset computation). However, SGD with momentum is used during the model's training (fresh) with the resulting coreset.
>
>
>
> **W3**. We also considered datasets with balanced label distribution in line with the long list of existing coreset works. This motivates us to choose the same number of samples from each class toward the coreset. Park et al. [1] have developed a coreset selection method for re-labelling applications for datasets with label noise. The label noise results in a class imbalance dataset and the authors have shown that a class-balanced version of their algorithm performs better than the version of the algorithm that does not choose the same number of samples per class. Hence, as reckoned by the reviewer, having a class-balanced coreset may result in better performance for class-imbalanced original dataset.
>
> **Suggested changes**
>
>
> 1. We thank the reviewer for pointing out the mistake. We have modified the manuscript accordingly.
>
>
> 2. We thank the reviewer for suggesting an alternate and better way to present the results. We have updated the manuscript accordingly.
>
>
>
>
> [1]  Park, Dongmin, et al. "Robust data pruning under label noise via maximizing re-labeling accuracy." Advances in Neural Information Processing Systems 36 (2024).

---

### Review · Reviewer_1AJm · 2025-02-17

**Summary Of Contributions:**

This paper investigates coreset selection, motivated by the observation that gradients of data within the same class exhibit strong cosine similarity. The authors propose selecting the top-ranked gradient cosine similarity within a class to represent the coreset. They validate the effectiveness of their method on image classification tasks using CIFAR-10, CIFAR-100, Tiny-ImageNet, and ImageNet-1k datasets, employing ResNet, VGG, and ViT architectures. Additionally, the method is tested on GANs for image generation, demonstrating superior performance over other data selection techniques. The paper also includes a detailed experimental analysis, highlighting the method's robustness against sample and label noise, as well as its generalization across different architectures.

**Audience:**

Yes

**Broader Impact Concerns:**

None.

**Claims And Evidence:**

Yes

**Requested Changes:**

The proposed method is well-motivated, supported by comprehensive experiments and analysis, and the paper is well-organized. However, I have a concern regarding the naming of "noise-free gradients." A more convincing explanation or additional experiments to substantiate this term would strengthen the paper.

**Strengths And Weaknesses:**

**Strengths:**
1. The proposed method is well-motivated and clearly explained, with experimental results demonstrating its effectiveness.

2. The authors conduct extensive experiments to validate the method's effectiveness, robustness to sample and label noise, generalization across architectures, and applicability to GANs for image generation.

3. The paper is well-structured and easy to follow.

**Weaknesses:** The term "noise-free gradients" is inadequately justified. Although the authors argue that existing coreset methods based on 'distinctness' or 'difficulty' favor samples that give rise to noisy loss gradients, this claim lacks experimental support or a clear explanation. Providing a more convincing rationale or additional experiments to demonstrate how prior methods introduce noise and why the proposed method avoids it would significantly strengthen the paper.

---

> ### Author Response · Authors · 2025-02-17
> **Response to Reviewer 1AJm**
>
> We thank the reviewer for their valuable time and feedback.
>
> The proposed corset selection is based on gradient similarity, which inherently favours samples that produce correlated gradients. In other words, the gradients of these images share strong similarities with the majority of samples belonging to the same class. That is the reason why we refer to these gradients as "noise-free".  We would like to clarify that the term "noise" here does not refer to contamination of the gradients but rather an indication that these images are the most representative of their particular class.
>
> We will update the manuscript to reflect this motivation and add more clarity.

---

> > ### Comment · Reviewer_1AJm · 2025-02-20
> > **Thanks for the authors' response**
> >
> > Thank you to the authors for addressing my concerns. After reviewing the response, my concern regarding the naming of the method has been resolved.

---

### Decision · Action_Editor_zqvk · 2025-04-18

**Recommendation:** Accept as is

**Comment:**

This paper proposes a simple, intuitive and computationally cheap method that works very well in the experiments. The authors also show an impressive number of ablation results, going beyond a standard evaluation in a typical paper. The reviewers were also unanimous in recommending acceptance, and all of the major questions they had were addressed by the authors.

**Audience:**

The paper is of interest to the TMLR community: it proposes a simple and effective method for coreset selection.

**Claims And Evidence:**

The paper studies the problem of coreset selection, i.e. selection of a small number of examples from a large dataset for training a model. The authors propose a simple idea: use datapoints which have gradients with high cosine similarity to other examples in the same class. They average the gradient over several checkpoints of the model. In the final version of the method, Noise-Free Loss Gradients, they use a nearest-neighbor inspired method for selecting examples which have a the highest number of examples in the same class with gradient cosine similarity above a threshold. The authors show that the proposed method works well, and outperforms other baselines for coreset selection. They also propose an efficient implementation of the method. They run multiple ablations showing that the method is robust to image corruptions and label noise, and also that it is robust to some hyper-parameter choices.

The claims made in the paper are well-supported with evidence. The authors report extensive experiment results across a large number of benchmark datasets. They also show an impressive number of additional ablation results, which further support their claims.